# Axon-Autonomous Effects of the Amyloid Precursor Protein Intracellular Domain (AICD) on Kinase Signaling and Fast Axonal Transport

**DOI:** 10.3390/cells12192403

**Published:** 2023-10-04

**Authors:** Svenja König, Nadine Schmidt, Karin Bechberger, Sarah Morris, Mercedes Priego, Hannah Zaky, Yuyu Song, Jan Pielage, Silke Brunholz, Scott T. Brady, Stefan Kins, Gerardo Morfini

**Affiliations:** 1Department for Human Biology and Human Genetics, University of Kaiserslautern-Landau, 67663 Kaiserslautern, Germanykarinbechberger@gmx.de (K.B.); kins@rptu.de (S.K.); 2Department of Anatomy and Cell Biology, University of Illinois at Chicago, Chicago, IL 60612, USAstbrady@uic.edu (S.T.B.); 3Marine Biological Laboratory, Woods Hole, MA 02543, USA; 4Department of Neurology, Harvard Medical School and Massachusetts General Hospital, Boston, MA 02129, USA; 5Department of Zoology, University of Kaiserslautern-Landau, 67663 Kaiserslautern, Germany; pielage@rptu.de

**Keywords:** Alzheimer’s disease, APP, AICD, kinases, NPTY

## Abstract

The amyloid precursor protein (APP) is a key molecular component of Alzheimer’s disease (AD) pathogenesis. Proteolytic APP processing generates various cleavage products, including extracellular amyloid beta (Aβ) and the cytoplasmic APP intracellular domain (AICD). Although the role of AICD in the activation of kinase signaling pathways is well established in the context of full-length APP, little is known about intracellular effects of the AICD fragment, particularly within discrete neuronal compartments. Deficits in fast axonal transport (FAT) and axonopathy documented in AD-affected neurons prompted us to evaluate potential axon-autonomous effects of the AICD fragment for the first time. Vesicle motility assays using the isolated squid axoplasm preparation revealed inhibition of FAT by AICD. Biochemical experiments linked this effect to aberrant activation of selected axonal kinases and heightened phosphorylation of the anterograde motor protein conventional kinesin, consistent with precedents showing phosphorylation-dependent regulation of motors proteins powering FAT. Pharmacological inhibitors of these kinases alleviated the AICD inhibitory effect on FAT. Deletion experiments indicated this effect requires a sequence encompassing the NPTY motif in AICD and interacting axonal proteins containing a phosphotyrosine-binding domain. Collectively, these results provide a proof of principle for axon-specific effects of AICD, further suggesting a potential mechanistic framework linking alterations in APP processing, FAT deficits, and axonal pathology in AD.

## 1. Introduction

Fast axonal transport (FAT), a cellular process involving long-range, bidirectional transport of membrane-bound organelles (MBOs) along axons, is driven by microtubule-based molecular motors (reviewed in [1]). In mature neurons, FAT in the anterograde and retrograde directions is mainly powered by the multi-subunit motor proteins: conventional kinesin (also known as kinesin-1) and cytoplasmic dynein, respectively [2]. The critical dependence of neurons on the maintenance of FAT is highlighted by a significant body of evidence. For example, mutations in genes that encode specific subunits of these motor proteins have been linked to some familial forms of neurodegenerative diseases, where affected neurons follow a dying-back pattern of degeneration [3,4]. Furthermore, many adult-onset neurodegenerative diseases, including Alzheimer’s disease (AD), feature alterations in FAT and exhibit dying-back degeneration of axons, despite not being associated with mutations in motor proteins. Mechanisms underlying such deficits have been a subject of debate, but several neuropathogenic proteins have been shown to elicit alterations in FAT by promoting aberrant activation of selected kinase pathways that in turn impact phosphorylation of motor proteins [3,5]. For example, mutant huntingtin, a protein that causes Huntington disease, promotes aberrant activation of JNK3, which in turn inhibits FAT by phosphorylating heavy-chain subunits of conventional kinesin [5]. Similarly, the ALS-related mutant proteins SOD1 and FUS inhibit FAT by promoting abnormal activation of p38 kinases [6,7,8]. Relevant to AD, the amyloid beta (Aβ) peptide produced by pathological processing of the amyloid precursor protein (APP) was shown to inhibit FAT by promoting activation of casein kinase 2 (CK2) [9].

Although a large body of evidence points to Aβ as a central component contributing to AD pathogenesis, proteolytic fragments other than Aβ are also produced as a result of APP processing, including the APP intracellular domain (AICD), which gets released into the cytoplasm [10,11]. AICD is produced by both the non-amyloidogenic processing pathway and, more prominently, the amyloidogenic processing pathway that leads to Aβ production [12,13], but the physiological effects of this cleavage are poorly understood. Notably, increased levels of both Aβ and AICD have been detected in the brains of patients affected by AD [10,14,15], with AICD playing a role in the heightened GluN2B synaptic function during aging, potentially underlying age-related synaptic impairments [16]. In line with this, transgenic expression of the AICD fragment in vivo was shown to promote AD-like pathological features in mice, including aberrant activation of kinases and abnormal patterns of neuronal protein phosphorylation [11,17,18,19]. Considering these precedents, and given the prominent role of protein kinases in the regulation of motor proteins [3,20], we hypothesized that increased AICD production might affect FAT, potentially contributing to AD pathogenesis [4,21,22].

To evaluate this hypothesis, recombinant forms of AICD were produced and their effects on FAT evaluated using vesicle motility assays in the isolated squid axoplasm preparation, as described before with other unrelated neuropathogenic proteins [3,23]. Collectively, results from these experiments revealed that AICD inhibits both anterograde and, to a lesser extent, retrograde FAT. Biochemical and pharmacological experiments further linked this effect to abnormal activation of selected axonal kinases and heightened phosphorylation of conventional kinesin, a major multi-subunit motor protein responsible for anterograde FAT. In addition, both deletion and competitive inhibition experiments suggest that interactions of one or more phosphotyrosine-binding domain (PTB)-containing axonal proteins with the NPTY motif in AICD might play a crucial role within this mechanism.

## 2. Materials and Methods

### 2.1. Recombinant GST–AICD Proteins

Recombinant cDNAs encoding AICD and mutants lacking the NPTY, YTSI, or PEER motif [24] were subcloned into the pGEX 4T2 vector. *Escherichia coli* (BL21) were transformed with these plasmids, and protein expression was induced for 2.5 h by addition of 1mM isopropyl β-D-1-thiogalactopyranoside (IPTG). Bacteria were harvested and resuspended in PBS with lysozyme (30 µg/mL). Bacterial lysates were sonicated, and then, 1% Triton X-100 (*v*/*v*) was added. After centrifugation (30 min, 1200× *g*, 4 °C), the insoluble debris was discarded and GST fusion proteins were purified from the supernatant using the Glutathione-Sepharose matrix (GE Healthcare, Solingen, Germany) according to the manufacturer’s protocol. After elution with glutathione-containing buffer, proteins were exchanged to 20 mM HEPES/150 mM NaCl/pH 7.2 using a PD-10 desalting column (Cytiva, Freiburg, Germany). Single-use aliquots of purified proteins were flash-frozen in liquid nitrogen immediately after preparation and stored at −80 °C until use.

### 2.2. Recombinant Fe65 PTB2-6xHis Protein

A plasmid-encoding recombinant, 6xHis-tagged PTB2 motif of the AICD-interacting protein partner Fe65 was expressed in *Escherichia coli* (BL21) using the plasmid vector pET21d Fe65-PTB2 6xHis [25]. Bacteria were cultivated at 37 °C in 2xYT media containing 100 mg/L of ampicillin. Protein expression was induced using 1 mM IPTG. After 20 h, the bacteria were sonicated 5 times with 10 pulses (Sonoplus HD 2200 sonicator, Bandelin, Germany) in lysis buffer (50 mM Tris/300 mM NaCl/10mM imidazole/pH 8) with EDTA-free protease inhibitor and 1mM DTT. Lysates were centrifuged for 45 min at 11,300× *g* at 4 °C, and proteins were purified using an Äkta Purifier 10 system (Cytiva) with a His-Trap HP column (GE Healthcare). After washing, proteins were eluted in 50 mM Tris/300 mM NaCl/300 mM imidazole/pH 8. Using a PD-10 desalting column (Cytiva), the elution buffer was exchanged to HEPES buffer (20 mM HEPES/150 mM NaCl, pH 7.2). The expression and purity of proteins were validated in Western blots using an anti-His antibody (Clontech # 631212). Single-use aliquots of purified proteins were flash-frozen in liquid nitrogen immediately after preparation and stored at −80 °C until use.

### 2.3. Squid Axoplasm Vesicle Motility Assay

Axoplasms from the squid giant axons (*Doryteuthis pealeii*; Marine Biological Laboratory, Woods Hole, MA, USA) were extruded, as described before [26]. Recombinant proteins (GST–AICD and deletion mutants) were diluted into 2 mM ATP-supplemented buffer X/2 (175 mM potassium aspartate, 65 mM taurine, 35 mM betaine, 25 mM glycine, 10 mM HEPES, 6.5 mM MgCl_2_, 5 mM EGTA, 1.5 mM CaCl_2_, 0.5 mM glucose, pH 7.2) and perfused with a final concentration of 100 nM. For kinase inhibition, the pharmacological inhibitors TBCA (2 µM; Calbiochem SML0854), SP600125 (500nM; Calbiochem S5567), SB203580 (5 µM; Calbiochem S8307), and ING135 (100 nM; [27,28]) were co-perfused along with GST–AICD proteins. FAT rates of membrane-bound vesicular organelles moving in anterograde and retrograde directions were examined by live imaging using a Zeiss Axiomat microscope equipped with a 100×, 1.3 N.A. objective and video differential interference contrast optics. Images were acquired using a Model C2400 CCD through a Hamamatsu Argus 20 processor and a Hamamatsu Photonics Microscopy C2117 video manipulator used to determine the flow velocity. Individual anterograde and retrograde FAT rates were obtained by matching their speed to a calibrated moving cursor over 50 min, as described before [26]. To determine mean FAT rates for each experimental condition, FAT rates recorded during the last 20 min of each individual experiment were pooled and analyzed [26].

### 2.4. Immunoprecipitation of P^32^-Radiolabeled Conventional Kinesin

Metabolic labeling experiments in isolated axoplasms were conducted, as described before [6,29]. Briefly, “sister” axons (dissected from the same squid) were extruded and perfused with Buffer X/2 containing radiolabeled P^32^-γ ATP and either 500 nM GST–AICD or GST. After 50 min incubation, axoplasms were lysed by adding 50 µL ROLB lysis buffer (50 mM HEPES, 100 mM NaCl, 0.5% (*v*/*v*) Triton X-100, 0.1% (*w*/*v*) sodium dehydrocholate, 50 mM sodium fluoride, 80 mM β-glycerophophate, 1 mM EDTA, pH 7.4) containing phosphatase inhibitors (2 mM sodium orthovanadate, 100 nM okadaic acid, 100 nM microcystin, phosphatase inhibitor cocktail (1:200; Calbiochem, Burlington, MA, USA), and 1:200 protease inhibitor cocktail (1:100 Sigma, Darmstadt, Germany)). Axoplasm lysates were collected from coverslips and homogenized with repeated pipetting, and an additional 50 µL of ROLB lysis buffer was added. Following this step, lysates were clarified using centrifugation and supernatants transferred to a fresh tube. An aliquot (3%) of clarified lysate was separated to ensure similar levels of radiolabeled proteins. Radiolabeled conventional kinesin was immunoprecipitated using an anti-KHC antibody (H2; [30]) and protein-G-conjugated protein beads (BioRad, Neuried, Germany). The immunoprecipitated material was collected using centrifugation and washed extensively with 50 mM HEPES, 200 mM NaCl, 0.5% (*w*/*v*) Triton X-100, pH 7.4. SDS sample buffer was added to both input lysates and immunoprecipitated proteins, and proteins were separated using SDS-PAGE (NuPAGE Novex 4–12% Bis-Tris gels; Invitrogen, Schwerte, Germany). Gels were dried and exposed to X-ray film for 0.5–20 h at −80 °C using an autoradiography cassette. Radiolabeled bands corresponding to KHC and KLCs were quantified using Image J ver 1.53k software and normalized to a radiolabeled band in the input lysate [29].

### 2.5. Immunoblot-Based Analysis of Kinase Activity

Two “sister” giant axons were dissected from individual squid and extruded onto glass slides, as described before [29]. For each axoplasm pair, one axoplasm was perfused with 30 µL of control perfusion mix (Buffer X/2) and the other with experimental perfusion mix (Buffer X/2 plus 500 nM GST-AICD). After 50 min incubation, the axoplasms were collected in 30 µL of 1% SDS. A 60 µL volume of 6-fold sample buffer was added to the lysates before separation on SDS-PAGE (NuPAGE Novex 4–12% Bis-Tris Gels; Invitrogen). Proteins were transferred to a PVDF membrane (BioRad) using Towbin buffer with 10% methanol. Membranes were rinsed and blocked in 1% nonfat dry milk (Nestle) diluted in TBS and supplemented with 2 mM sodium orthovanadate and 10 mM sodium fluoride to block contaminating phosphatases in milk. The following antibodies were used for Western blots: anti-KHC (H2 clone [30]), anti-p-p38 (Cell Signaling Technology #9215 lot#7, rabbit monoclonal), anti-p-JNK (Cell Signaling Technology #4668 lot#24, rabbit monoclonal), and anti-dp-ser9/21-GSK3α/β (clone 15C2; [31]). Primary and secondary antibodies were diluted in TBS containing 1% BSA and phosphatase inhibitors. Membranes were incubated with primary antibodies overnight at 4 °C. After washing, the membranes were incubated with secondary antibodies (Li-COR) for 1 h at RT, washed again, and imaged using a Li-COR Odyssey infrared imaging system. Li-COR ImageStudioLite 5.2 software was used to quantify the immunoreactivity of bands.

### 2.6. Statistics

Prism software (Graphpad 9.5) was used for statistical analysis of results. The Kolmogorov–Smirnov test was used to test for normal distribution. Data with a normal distribution were analyzed using an unpaired Student’s *t*-test. If the groups did not follow a normal distribution, the Mann–Whitney U-test was applied. Quantitative data in all graphs are represented as the mean ± standard error of the mean (SEM). Summaries of mean FAT rate values and statistical comparisons for all experimental conditions in this work are detailed in Tables 1 and 2, respectively.

## 3. Results

### 3.1. AICD Inhibits Anterograde and Retrograde FAT, Increasing also Phosphorylation of the Motor Protein Conventional Kinesin

Alterations in APP processing and early degeneration of axons have long been recognized in AD [4,32]. Despite this knowledge, specific physiological effects of APP fragments elicited in the axonal compartment remain undefined [33].

Biochemical and genetic interaction screens identified a wide variety of extracellular and intracellular APP-binding partners. AICD harbors several well-defined peptide motifs, which are interaction sites for proteins involved in the transduction of various signaling pathways ([10,34]; Figure 1A). These motifs raised the question of whether the AICD fragment might impact critical cellular processes sustaining axonal health, including FAT, as previously observed with the Aβ peptide [9] and disease-related forms of tau [35,36]. To address this possibility, potential effects of AICD on FAT were evaluated using vesicle motility assays in isolated squid axoplasms, a unique ex vivo preparation to study molecular events in the axonal compartment [26]. The absence of a plasma membrane in this preparation allows a direct evaluation of the effects triggered by neuropathogenic proteins on FAT and the identification of mechanisms and specific molecular components underlying such effects [5,7,9,36,37].

As glutathione-S-transferase (GST) does not elicit any effects on FAT [5], we prepared recombinant GST-tagged forms of AICD, as described before [24]. The purity of GST and GST–AICD proteins was validated using Coomassie gel staining (Figure 1B). Axoplasms were perfused with either Buffer X/2 alone as a control (Figure 2A, top panel) or with Buffer X/2 containing 100 nM GST–AICD (Figure 2A, bottom panel). FAT rates of vesicles moving in anterograde and retrograde directions were measured using video-enhanced contrast-differential interference contrast (VEC-DIC) over a period of 50 min [26]. In control axoplasms perfused with Buffer X/2, mean anterograde and retrograde FAT rates remained steady over the course of the experiment (1.64 ± 0.01 µm/s and 1.20 ± 0.03 µm/s, respectively) (Figure 2A, top panel). In contrast, a marked reduction in mean anterograde (1.23 ± 0.02 µm/s) and retrograde (1.02 ± 0.03 µm/s) FAT rates was recorded in axoplasms perfused with GST–AICD (Figure 2A, bottom panel). Statistical analysis of all individual FAT rate measurements obtained 30–50 min after perfusion of axoplasms with either Buffer X/2 or GST–AICD confirmed these findings (Figure 2B; Table 1 and Table 2). These data indicated that independently of any alterations in the neuronal cell body, AICD promotes deficits in both anterograde and retrograde FAT.

A growing body of evidence documenting FAT regulation by protein kinases raised the question of whether AICD might inhibit FAT by promoting alterations in the phosphorylation of motor proteins [3,20]. Metabolic labeling experiments were used to address this possibility, as described before for other neuropathogenic proteins [6,29]. Two “sister” axons (dissected from the same squid) were perfused with Buffer X/2 containing radiolabeled ATP and either GST or GST–AICD. Conventional kinesin was immunoprecipitated from radiolabeled axoplasm lysates using an antibody that recognizes kinesin heavy-chain subunits (H2 antibody; [38]). After separation of immunoprecipitates using SDS-PAGE, gels were dried and radiolabeled bands corresponding to kinesin heavy- (KHC) and light-chain (KLC) subunits of conventional kinesin identified with autoradiography (Figure 2C). Quantitative densitometric analysis revealed significant increases in the phosphorylation of both KHC (~31 ± 10%) and KLC (~23 ± 9%) subunits in axoplasms incubated with GST–AICD compared to those incubated with GST. Collectively, data in Figure 2C linked the inhibitory effects of AICD on FAT to aberrant phosphorylation of conventional kinesin—the main motor protein powering anterograde FAT in the isolated squid axoplasm model [38].

### 3.2. AICD Activates Selected Axonal Kinases

Increased phosphorylation of conventional kinesin following perfusion of axoplasms with GST–AICD strongly suggested activation of endogenous axonal squid kinases. Several kinases aberrantly activated in various neurodegenerative diseases have been shown to affect FAT through phosphorylation of motor proteins [3,20,23]. To evaluate potential activation of kinases already known to inhibit FAT by AICD, we performed immunoblotting experiments in “sister” axoplasms perfused with 500 nM of either GST (control) or GST–AICD proteins (Figure 3A). After 50 min incubation, axoplasms were lysed, proteins separated using SDS-PAGE, and immunoblots developed using antibodies that recognize phosphorylated, catalytically active forms of the MAPKs p38 and JNK (p-p38 and p-JNK, respectively) as well as an antibody that selectively recognizes active forms of the kinases GSK3α and GSK3β (dpGSK3; GSK3 dephosphorylated at the regulatory serine 9 residue; [31]). H2, a phosphorylation-independent antibody against KHC subunits [38], provided an internal control for protein loading (Figure 3B). Quantitative analysis of p-p38/KHC, p-JNK/KHC, and dpGSK3/KHC immunoreactivity ratios revealed increased levels of activated p38 and JNK, but not GSK3, in axoplasms perfused with GST–AICD compared to those perfused with GST (Figure 3C). Along with data in Figure 2C, these results indicate that AICD promotes activation of selected protein kinases in axons.

### 3.3. Inhibitory Effects of AICD on FAT Involve Activation of Specific MAPKs and CK2

Next, co-perfusion experiments were used to determine whether inhibitory effects of AICD on FAT were mediated by axonal kinases identified in Figure 3 and Figure 4. Toward this, GST–AICD was co-perfused with the pharmacological p38/JNK3 inhibitor SB203580 (5 µM, Figure 4A), the pan-JNK inhibitor SP600125 (500 nM, Figure 4B), or the GSK3 inhibitor ING-35 (100 nM, Figure 4C), which were all shown to inhibit their target kinases in the squid axoplasm preparation [5,6]. Based on our prior findings showing casein kinase 2 (CK2)-mediated inhibition of FAT by Aβ [9], AICD was also co-perfused with TBCA, a highly specific CK2 inhibitor previously tested in the axoplasm model (2 µM, Figure 4D; [39]). Inhibition of p38, JNK, and CK2 significantly prevented the reduction in mean anterograde FAT rate values elicited by GST–AICD (1.2 ± 0.03 µm/s; see Figure 2), albeit to different degrees (GST–AICD + SP600125 = 1.51 ± 0.01 µm/s; GST–AICD + SB203580 = 1.50 ± 0.01 µm/s; and GST–AICD + TBCA = 1.42 ± 0.03 µm/s; Figure 4E, left panel, and Table 1 and Table 2). Overall, p38, JNK, and CK2 inhibitors maintained mean anterograde FAT rates to levels close to controls (1.64 ± 0.01 µm/s). In contrast, mean anterograde FAT rates of axoplasms co-perfused with GST–AICD and the GSK3 inhibitor ING-35 (1.28 ± 0.01 µm/s) were similar to those observed in axoplasms perfused with GST–AICD alone (1.20 ± 0.03 µm/s). Concomitantly, the inhibitory effect of AICD on mean retrograde FAT rates (1.02 ± 0.03 µm/s; Figure 2) was also significantly prevented by p38, JNK, and CK2 inhibitors (GST-AICD + SB203580 = 1.22 ± 0.01 µm/s; GST-AICD + SP600125 = 1.15 ± 0.01 µm/s; and GST-AICD + TBCA = 1.12 ± 0.01 µm/s; Figure 4E, right panel; Table 1 and Table 2). Taken together, these experiments indicated that the inhibitory effect of AICD on FAT involves activation of several axonal kinases, including JNK, p38, and CK2.

**Table 2 cells-12-02403-t002:** Statistical comparison of results from vesicle motility assays (n = number of experiments; n.s. = not significant).

Figures Featuring Data	Experimental Comparisons (vs)		*p* Value	
			Anterograde	Retrograde
Figure 2	Buffer X/2 (control) (n = 7)	AICD (n = 7)	<0.0001	<0.0001
Figure 4	AICD (n = 7)	AICD + SB203580 (n = 3)	<0.0001	<0.0001
	AICD (n = 7)	AICD + SP600125 (n = 3)	<0.0001	<0.0001
	AICD (n = 7)	AICD + ING35 (n = 3)	0.0036	<0.0001
	AICD (n = 7)	AICD + TBCA (n = 3)	<0.0001	0.0009
	Buffer (n = 7)	AICD + SB203580 (n = 3)	<0.0001	n.s. (0.7361)
	Buffer (n = 7)	AICD + SP600125 (n = 3)	<0.0001	0.0095
	Buffer (n = 7)	AICD + ING35 (n = 3)	<0.0001	0.0163
	Buffer (n = 7)	AICD + TBCA (n = 3)	<0.0001	0.0011
Figure 5	AICD (n = 7)	AICDΔYTSI (n = 3)	<0.0001	<0.0001
	AICD (n = 7)	AICDΔPEER (n = 2)	0.0005	<0.0001
	AICD (n = 7)	AICDΔNPTY (n = 6)	<0.0001	<0.0001
	Buffer (n = 7)	AICDΔYTSI (n = 3)	<0.0001	<0.0001
	Buffer (n = 7)	AICDΔPEER (n = 2)	<0.0001	<0.0001
	Buffer (n = 7)	AICDΔNPTY (n = 6)	<0.0001	n.s. (0.0026)
Figure 6	AICD (n = 7)	AICD + PTB2-His (n = 5)	<0.0001	<0.0001
	Buffer (n = 7)	AICD + PTB2-His (n = 5)	<0.0001	n.s. (0.2272)

### 3.4. The Inhibitory Effect of AICD on FAT Depends on a Peptide Sequence Encompassing the NPTY Motif

To map specific peptide motifs in AICD mediating its inhibitory effect on FAT, deletion experiments were performed, as described before with other pathogenic proteins [36,39]. Toward this, GST-tagged AICD proteins lacking peptide sequences that encompass the PEER (GST-AICDΔPEER)-, YTSI (GST-AICDΔYTSI)-, and NPTY (GST-AICDΔNPTY)-binding motifs were purified (Figure 5A,B), diluted in Buffer X/2 at 100 nM, and perfused in axoplasms for vesicle motility assays. Perfusion of either GST–AICDΔPEER (Figure 5C) or GST–AICDΔYTSI (Figure 5D) inhibited FAT rates in both directions. In contrast, both anterograde and retrograde FAT rates remained constant throughout the course of the experiments following GST–AICDΔNPTY perfusion (Figure 5D). Quantitative analysis indicated that mean anterograde FAT rates were reduced in axoplasms perfused with either GST–AICDΔPEER (1.08 ± 0.02 µm/s) or GST–AICDΔYTSI (0.97 ± 0.02 µm/s) compared to those perfused with GST–AICD (1.2 ± 0.03 µm/s) (Figure 5F, top panel, Table 1 and Table 2). Similar results were observed for mean retrograde FAT rates (Figure 5F, bottom panel, Table 1 and Table 2). In contrast, effects of GST–AICDΔNPTY perfusion on mean FAT rates were minimal, with mean anterograde (1.49 ± 0.03 µm/s) and retrograde (1.14 ± 0.01 µm/s) FAT rates close to those observed in control axoplasms perfused with Buffer X/2 (1.64 µm/s and 1.23 µm/s, respectively; Figure 2A). Collectively, these data strongly suggested that the NPTY motif might play a crucial role in mediating the inhibitory effect of AICD on FAT. In addition, GST–AICDΔYTSI and GST–AICDΔPEER inhibited anterograde FAT rates to a larger extent compared to GST–AICD, which may reflect increased exposure of the NPTY motif in proteins with deletion of YTSI or PEER.

### 3.5. Inhibitory Effects of AICD on FAT Involve NPTY Motif Interactions with Endogenous PTB-Containing Squid Proteins

The NPTY motif in AICD interacts with different scaffolding proteins, including Fe65 and JIP1/2, that in turn interact with and coordinate the activation of numerous protein kinases [10,19] (Figure 1A). This prompted us to consider whether the effects of AICD on axonal kinases and FAT could be mediated by one or more endogenous squid protein ligands containing a PTB motif. The participation of a PTB-containing protein on the inhibitory effects elicited by GST–AICD was tested using competitive inhibition experiments (Figure 6A). Toward this, a recombinant, 6x-His-tagged form of the Fe65 PTB2 domain (PTB2-6xHis), known to interact with the NPTY motif in AICD [25,40], was purified and validated with immunoblot analysis using an anti-His antibody (Figure 6B). Next, GST–AICD (100 nM) was co-perfused with PTB2 in a molar excess (500 mM) to prevent potential NPTY interactions with endogenous PTB-containing squid proteins. Consistent with the mild effects of GST–AICDΔNPTY on FAT (Figure 5E), the inhibitory effect of GST–AICD on FAT was markedly attenuated by PTB2-6xHis co-perfusion (Figure 6C). Quantitative analysis of mean FAT rates revealed that PTB2-6xHis significantly prevented the inhibitory effect of AICD on both anterograde (GST–AICD = 1.20 ± 0.03 µm/s versus GST–AICD + PTPB2-6xHis = 1.40 ± 0.02 µm/s) and retrograde (GST–AICD = 1.02 ± 0.03 µm/s versus GST–AICD + PTPB2-6xHis = 1.19 ± 0.02 µm/s) FAT (Figure 6D, Table 1 and Table 2). These data indicated that inhibition of FAT by AICD involves, at least in part, interactions of the NPTY motif with one or more endogenous squid proteins containing phosphotyrosine-binding motif(s).

## 4. Discussion

### 4.1. AICD Impacts FAT via Activation of Axonal Kinases

Both Aβ and AICD are produced during the proteolytic cleavage of APP by γ-secretase [12,13]. Aberrant Aβ production gives rise to the stereotypic amyloid plaques found in AD brains [10]. Interestingly, some studies have also reported elevated AICD levels in AD-affected brain tissues [41], suggesting this APP fragment could also contribute to AD pathogenesis. In support, AICD overexpression in vivo has been associated with increased activation of kinases that, in turn, increased tau phosphorylation [17,19,41]. Consistent with these precedents, data in this study provide compelling evidence showing that AICD can promote the activation of axonal kinases that, in turn, inhibit FAT by affecting the phosphorylation of motor proteins. Pharmacological and biochemical data suggested that MAPKs and CK2 contribute to the inhibitory effects of AICD on FAT (Figure 3 and Figure 4). Activation of these kinases was associated with increased phosphorylation of KHC subunits of conventional kinesin, suggesting that AICD may also promote the phosphorylation of additional axonal proteins, such as tau [17,19,41]. These data are also in line with previous reports showing that JNK, p38, and CK2 inhibit FAT by phosphorylating conventional kinesin [3,5,6,8,23,37,42].

Interestingly, our data show that axonal GSK3 is not activated by AICD, nor does this kinase mediate its inhibitory effect on FAT. This is of particular interest, as prior studies have documented GSK3β activation in AICD-overexpressing mice [17,18,43] and AD-affected brain tissues [3]. One possible explanation for these different effects of AICD on different pathways in distinct experimental settings is that some effects are restricted or markedly more prominent on specific subcellular compartments. For example, AICD might predominantly activate MAP kinase and CK2 pathways in axons, whereas it might activate a different subset of kinases, including GSK3β, in the cell soma or in dendrites. Alternatively, previous reports that GSK3β was activated by AICD may represent an indirect effect. For example, phosphorylation of tau at specific sites can promote activation of a phosphatase-dependent pathway leading to GSK3 activation [36,44,45,46]. Studies showing AICD-dependent activation of GSK3β or protein kinase A (PKA) all relied on analysis of whole-brain or whole-cell lysates [16,17,18,43,47]. In those studies, the impact of AICD was reported to depend on kinase activation as well as nuclear signaling, and modulation of NMDAR and L-type Ca^2+^ channels, which are lacking in the axoplasm [15,16,48,49,50,51]. Together, these data strongly indicate that not only the increased production but also the subcellular compartment where the AICD fragment is generated could affect its pathophysiological function. 

Numerous independent observations have implicated the NPTY motif in the biological effects of AICD. Previous studies have shown interactions of the NPTY motif in AICD with various PTB proteins, including Fe65, X11/Mint family members, Numb, and JIP1/2 [10,52]. Consistent with this, our study showed that the inhibitory effect of AICD on FAT could be counteracted by an excess of the Fe65 PTB2 domain (Figure 6), which exhibits high affinity for the APP NPTY motif [53]. JIP1/2, which affects the MAP kinase pathway through JNK activation, also interacts with the cytoplasmic domain of the APP NPTY motif [54,55,56]. Investigations of axonopathies resulting from interactions between PTB proteins and the NPTY motif highlighted the role of JIP1 [57]. While these studies were conducted in the *Drosophila* fly model, conservation of molecular mechanisms for regulation of kinases has also been demonstrated in other models, including squid. This assumption is supported by data suggesting that JNK-interacting protein (JIP1) mediates AD-like pathologies in AICD-overexpressing mice, most probably through JNK activation [19]. However, considering the highly complex composition of the AICD interactome [11], it is conceivable that additional indirect effects, beyond the interaction of AICD with JIP and JNK, may facilitate the activation of the MAP kinase pathway. 

A novel finding in this study was the observation that pharmacological inhibition of CK2 partially rescued AICD’s inhibitory effect on FAT (Figure 4). CK2 is implicated in various human diseases, including Parkinson’s and AD, and is activated by Aβ oligomers [9]. This led to the suggestion that CK2 may be a potential therapeutic target in AD [58]. Pharmacological data in this work indicate that CK2 may contribute to the observed inhibition of FAT by AICD. However, since inhibition of the MAP kinase pathway through p38 and JNK inhibitors was sufficient to restore anterograde FAT during AICD perfusion, the effects of CK2 may be due to actions downstream of the MAP kinase pathway. Supporting this possibility, prior studies have reported a link between MAPK and CK2 activation [59,60]. In summary, our study provides compelling evidence that AICD promotes NPTY-motif-dependent activation of selected axonal kinases, which in turn inhibit FAT through a mechanism involving, in part, phosphorylation of conventional kinesin [2].

### 4.2. AICD’s Potential Involvement in AD-Linked Axonopathy

A substantial amount of evidence indicates that neurons affected in AD patients undergo a degenerative process known as “dying-back axonopathy”. This means that abnormalities in synaptic function and axonal connectivity develop long before the death and loss of neuronal cell bodies. The specific mechanisms responsible for the dying-back degeneration of neurons in AD are still not fully defined, but various hypotheses have been put forward, including deficits in FAT [4], signaling abnormalities elicited by Aβ and Tau [61,62], and Aβ and Tau aggregation, which has also been linked to FAT inhibition [9,23,36]. For example, pathological forms of tau that lead to a conformational change in tau, whether due to phosphorylation, mutation, or aggregation [35,62], lead to protein phosphatase 1 (PP1)-dependent activation of GSK3, which in turn inhibits anterograde FAT [23,28,36,45]. Similarly, perfusion of Aβ into the axoplasm causes CK2 activation, which leads to inhibition of both retrograde and anterograde FAT [9].

Evidence has accumulated suggesting that AICD, the primary cytosolic released fragment of APP, is increased under pathological conditions in the brain of AD patients [41,49,63]. Moreover, AICD transgenic mice exhibit AD-like features, including abnormal phosphorylation of tau [41], and APP overexpression in neuronal cells, which resulted in axonopathies, was linked to activation of the MAPK pathway by AICD in a NPTY-motif-dependent manner [18,57]. Collectively, these findings suggest that increased AICD levels early in AD could contribute to the development of axonopathies. Consistent with this notion, our data provide compelling evidence indicating that in the axonal compartment, AICD can affect FAT by promoting the activation of MAP kinase pathways and CK2. Disruption of kinase signaling in neurons may also have additional effects. For example, studies using whole-brain or whole-cell lysates, which do not distinguish between effects in soma and cellular processes or terminals, described additional kinases that were activated by transgenic expression of AICD [17,18,43]. This may reflect the different molecular composition and environment of specific neuronal subcompartments other than axons, where AICD would disrupt other pathways. Consistent with this view, it was recently reported that APP internalization in the neuronal somatodendritic compartment differs from that in the axonal compartment, being clathrin independent. Further, this report also suggested that the somatodendritic APP pool does not represent a major source of Aβ generation, indicating that neurons mainly compartmentalize surface APP pools toward downstream processing pathways in axons [64]. Alternatively, indirect effects of AICD, including aberrant phosphorylation of tau, can lead to GSK3β activation in these whole-brain studies [17,36,45]. The results presented here favor the hypothesis that in axons, AICD preferentially activates the MAPK and CK2 pathways, both of which are known to phosphorylate tau and conventional kinesin [5,6,9]. 

The idea of AICD as a contributor to AD pathology gets further support from a study showing that the familial AD (fAD)-related mutant protein APPswe results in increased AICD production [18] and that other fAD mutations increase both Aβ levels and the generation of AICD [65]. The available evidence suggests that a diverse set of neuropathogenic proteins/peptides, including Aβ and tau, induce alterations in FAT by promoting the activation of selected protein kinases involved in the regulation of molecular motors (reviewed in [3]). The complex changes in neuronal signaling pathways in AD pathogenesis make it likely that AICD synergistically affects FAT similarly, as suggested previously for the GSK3 hypothesis. 

## 5. Conclusions

Our data clearly implicate increases in AICD levels as a potential pathogenic factor contributing to FAT deficits in AD. We identified toxic, axon-autonomous effects of this APP fragment on FAT that involve interactions of the NPTY motif with one or more phosphotyrosine-binding domain-containing proteins, as well as abnormal activation of protein kinases and aberrant phosphorylation of conventional kinesin. Given the unique dependence of axons on sustained functionality of FAT, AICD should be considered as a potential pathogenic factor contributing to axonal degeneration in AD.

## Figures and Tables

**Figure 1 cells-12-02403-f001:**
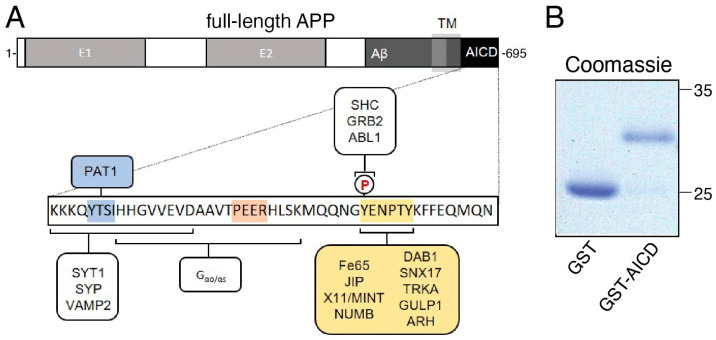
APP interacts with numerous protein ligands through specific motifs. (**A**) Top: schematic illustration of full-length amyloid precursor protein (APP) showing its long extracellular as well as its short intracellular domain (AICD). Bottom: selected interaction partners of the APP intracellular domain (AICD) and specific peptide motifs involved are highlighted. (**B**) A cDNA encoding AICD fused to glutathione-S-transferase (GST–AICD) was used to express and purify recombinant protein from *E. coli*. Purified GST and GST–AICD proteins were separated using SDS-PAGE and stained with Coomassie Blue. Abbreviations: TM = transmembrane domain; PAT1 = protein interacting with APP tail 1; SYT1 = synaptotagmin 1; SYP = synaptophysin; VAMP2 = vesicle-associated membrane protein 2; SHC = SRC homology 2 domain-containing-transforming protein 1; GRB2 = growth factor receptor-bound protein 2; JIP = cJun N-terminal kinase-interacting protein; DAB1 = disabled homologue 1; SNX17 = sorting nexin 17; TRKA = tyrosine kinase receptor A; GULP1 = engulfment adapter PTB domain containing 1; ARH = low-density lipoprotein receptor adaptor protein 1 (LDLRAP1).

**Figure 2 cells-12-02403-f002:**
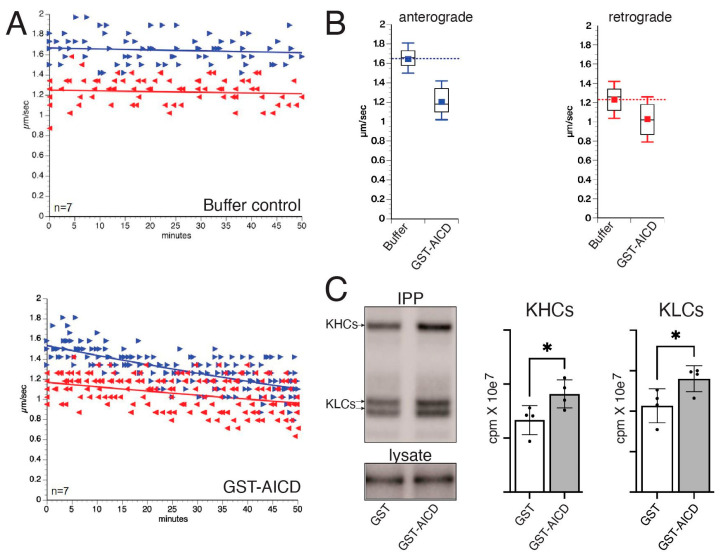
AICD inhibits fast axonal transport (FAT), an effect that correlated with increased phosphorylation of conventional kinesin. (**A**) Vesicle motility assays in isolated squid axoplasm showing the effects of Buffer X/2 alone (control, top) or Buffer X/2 mixed with 100 nM GST–AICD (bottom). Both anterograde and retrograde FAT rates were obtained using video-enhanced contrast-differential interference contrast microscopy over a period of 50 min. Graphs were plotted as a function of velocity (µm/s) and time. Right blue arrows (►) correspond to individual anterograde FAT rate measurements, whereas left red arrows (◄) indicate individual retrograde FAT rates. FAT rates remained constant in control axoplasms perfused with Buffer X/2 (buffer control, top panel). In contrast, a time-dependent reduction in both anterograde and retrograde FAT rates was observed in axoplasms perfused with GST–AICD (lower panel). (**B**) Quantitation of individual anterograde (left panel) and retrograde (right panel) FAT rate measurements collected between 30 and 50 min after perfusion. For reference, dashed blue (anterograde) and red (retrograde) lines indicate mean FAT values for control axoplasms perfused with Buffer X/2 (1.64 µm/s and 1.23 µm/s, respectively) (**C**) Axoplasm pairs (see Methods) were perfused with 1 mM radiolabeled P^32^ ATP and 500 nM GST or GST–AICD (n = 4 each). Following lysis, conventional kinesin was immunoprecipitated using an anti-kinesin-1 (H2) antibody, and immunoprecipitates (IPP) were separated using SDS-PAGE. Radiolabeled bands corresponding to heavy chains (KHC) and light chains (KLC) of conventional kinesin are shown (upper panel; IPP) [6,29]. As a loading control, 3% of the input axoplasm lysates prior to IPP were loaded and separated using SDS-PAGE. One of the numerous phosphorylated proteins in input lysates was used for normalization (lower panel, lysate). Densitometric quantification of autoradiograms showed increased P^32^ incorporation in both KHC and KLC subunits in lysates prepared from axoplasms perfused with GST–AICD compared to axoplasms perfused with GST. * *p* = 0.05.

**Figure 3 cells-12-02403-f003:**
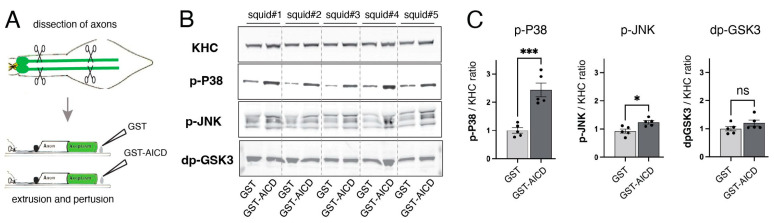
AICD activates axonal p38 and JNK. (**A**) Two “sister” giant axon pairs were dissected from five squid. Each “sister” axon pair was perfused with either 500 nM GST or 500 nM GST–AICD, both diluted in Buffer X/2. Axoplasm lysates were prepared, separated using SDS-PAGE, and processed for immunoblotting. (**B**) Immunoblots were developed with antibodies that recognize active (activation loop-phosphorylated) forms of p38 and JNK (p-p38 and p-JNK, respectively), as well with an antibody that recognizes dephosphorylated (dp, active) forms of GSK3α and GSK3β [31]. An antibody recognizing kinesin heavy-chain subunits (KHC) provided a control for total axoplasmic protein loading (H2 antibody; [38]). (**C**) Plots depicting LICOR-based quantitation of anti-p-p38, anti-p-JNK, and anti-dpGSK3 immunoreactivities, normalized to that from anti-KHC antibody. Perfusion of GST–AICD significantly increased levels of active p38 (p-p38) and JNK (pJNK) compared to sister axoplasms perfused with GST. In contrast, levels of active (serine9-dephosphorylated) GSK3 (dp-GSK3) remained unchanged. *** *p* = 0.001, * *p* = 0.05, ns = not significant; n = 5.

**Figure 4 cells-12-02403-f004:**
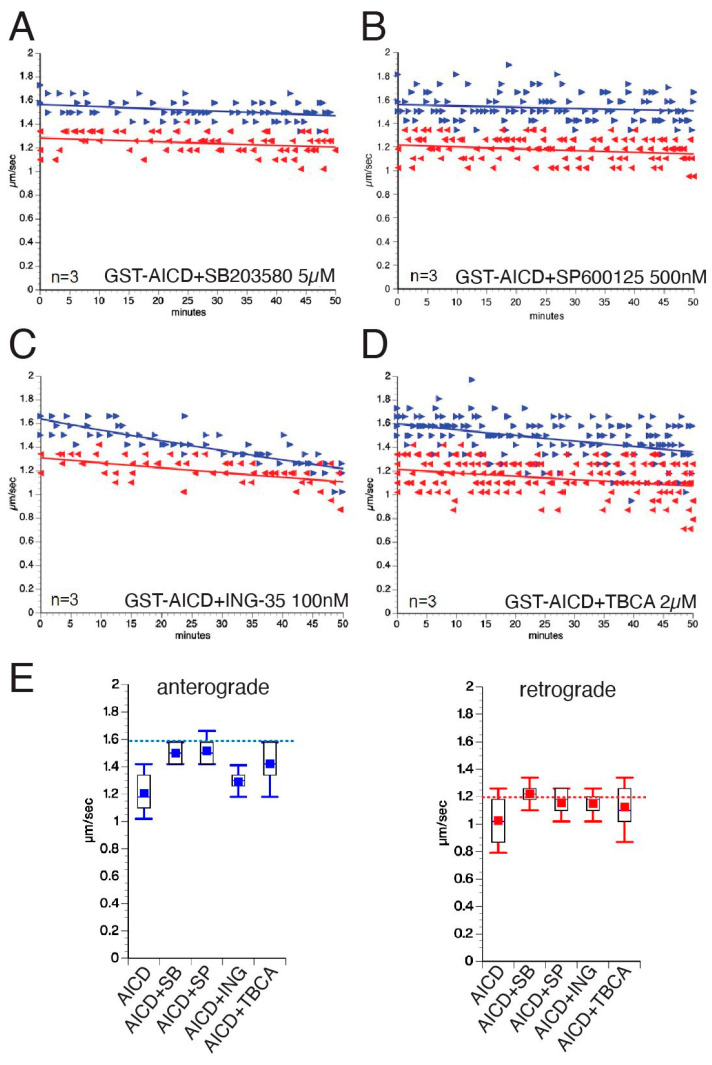
p38, JNK, and CK2 contribute to the inhibitory effect of AICD on fast axonal transport. (**A**–**D**) Vesicle motility assays in isolated squid axoplasm. Isolated axoplasms were co-perfused with 100 nM GST–AICD and specific kinase inhibitors, including 5 µM SB203580 (p38 kinase inhibitor; (**A**)), 500 nM SP600125 (JNK inhibitor; (**B**)), 100 nM ING-135 (GSK3 inhibitor; (**C**)), and 2 µM TBCA (CK2 inhibitor; (**D**)). Anterograde (►, blue arrows) and retrograde (◄, red arrows) FAT rates were measured via video-enhanced contrast-differential interference contrast microscopy over 50 min. Graphs are plotted as velocity (µm/s) against time (minutes). (**E**) Quantitation of mean anterograde (left panel) and retrograde (right panel) FAT rates from experiments in panels (**A**–**D**) (see Table 1 and Table 2). For reference, dashed blue (anterograde) and red (retrograde) lines are shown indicating mean FAT values for control axoplasms perfused with Buffer X/2 (1.64 µm/s and 1.23 µm/s, respectively).

**Figure 5 cells-12-02403-f005:**
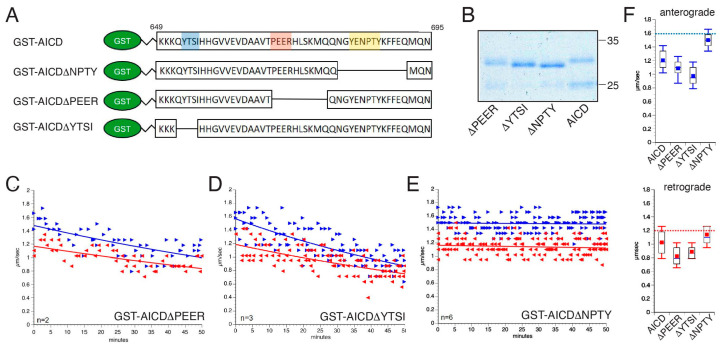
The inhibitory effect of AICD on FAT depends on a peptide sequence encompassing the NPTY motif. (**A**) Schematic illustration of glutathione-S-transferase (GST) fusion proteins coupled to either the APP intracellular domain (GST–AICD) or the GST–AICD mutants lacking the YTSI (blue), PEER (orange), or NPTY (yellow) motif. (**B**) Coomassie staining of affinity-purified GST fusion proteins in (**A**) after separation with SDS-PAGE. (**C**–**E**) Vesicle motility assays in isolated squid axoplasm. Isolated axoplasms were co-perfused with 100 nM GST–AICD or the indicated GST–AICD mutants (**C**) ΔPEER, (**D**) ΔYTSI, or (**E**) ΔNPTY. Anterograde (►, blue arrows) and retrograde (◄, red arrows) FAT rates were obtained using video-enhanced contrast-differential interference contrast microscopy over 50 min. Graphs are plotted as velocity (µm/s) against time. (**F**) Quantitation of mean anterograde (top panel) and retrograde (bottom panel) FAT rates from experiments in panels (**C**–**E**) compared to GST–AICD (see Table 1 and Table 2). For reference, dashed blue (anterograde) and red (retrograde) lines are shown indicating mean FAT values for control axoplasms perfused with Buffer X/2 (1.64 µm/s and 1.23 µm/s, respectively).

**Figure 6 cells-12-02403-f006:**
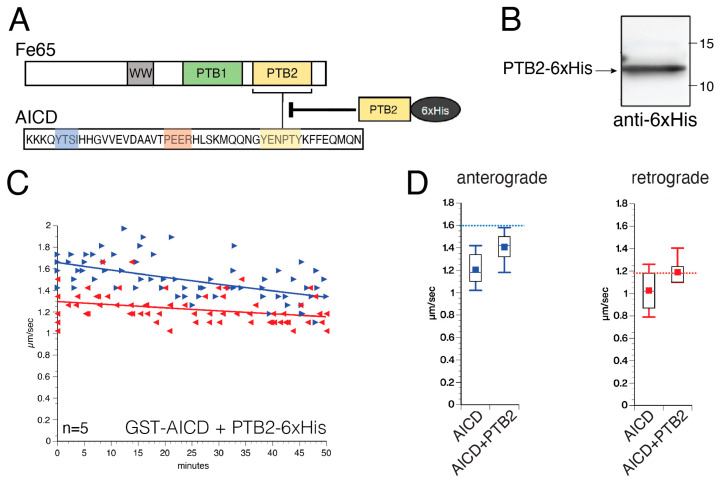
The inhibitory effect of AICD on FAT is significantly reduced by co-perfusion of a phosphotyrosine-binding (PTB) domain. (**A**) Schematic illustration of the APP interacting protein Fe65 containing different protein interaction domains, such as the WW, PTB1, and PTB2 domains. Highlighted in yellow is the PTB2 domain, which has been shown to interact with the NPTY motif in AICD. Perfused at higher molar concentrations, a recombinant version of a 6xHis-tagged PTB2 protein would be expected to compete with NPTY interactions with endogenous squid proteins containing a similar domain (inhibitory arrow). (**B**) Western blot of the purified recombinant PTB2-6xHis protein, developed using an anti-6xHis antibody. (**C**) Vesicle motility assay performed in isolated squid axoplasm showing the effect of 100 nM AICD co-perfused with 500 nM PTB2-6xHis. Anterograde (►, blue arrows) and retrograde (◄, red arrows) FAT rates were obtained via video-enhanced contrast-differential interference contrast microscopy over 50 min. Graphs are plotted as velocity (µm/s) against time. (**D**) Quantitation of mean FAT rates from experiments in (**C**). PTB2-6His co-perfusion significantly prevented the effects of AICD on both mean anterograde (left panel) and retrograde (right panel) FAT rates. For reference, dashed blue (anterograde) and red (retrograde) lines indicate mean FAT values for control axoplasms perfused with Buffer X/2.

**Table 1 cells-12-02403-t001:** Statistical analysis of fast axonal transport (FAT) rates recorded from vesicle motility assays (*n* = number of independent FAT rate measurements after 20 min of perfusion.

Experimental Condition	Direction	*n*	Mean	Variance	Standard Deviation	Standard Error
Buffer X/2 (control)	Anterograde	24	1.64	0.01	0.118743	0.024238
	Retrograde	27	1.23	0.01	0.126022	0.024253
AICD	Anterograde	81	1.20	0.03	0.158260	0.017584
	Retrograde	78	1.02	0.03	0.181814	0.020586
AICD + SB203580	Anterograde	33	1.50	0.01	0.073443	0.012785
	Retrograde	36	1.22	0.01	0.088644	0.014774
AICD + SP600125	Anterograde	52	1.51	0.01	0.107138	0.014857
	Retrograde	50	1.15	0.01	0.098633	0.013949
AICD + ING-35	Anterograde	26	1.28	0.01	0.107972	0.021175
	Retrograde	25	1.15	0.01	0.101575	0.020315
AICD + TBCA	Anterograde	65	1.42	0.03	0.166072	0.020599
	Retrograde	66	1.12	0.03	0.162290	0.019977
AICDΔPEER	Anterograde	23	1.08	0.02	0.124697	0.026001
	Retrograde	22	0.89	0.01	0.091746	0.019560
AICDΔYTSI	Anterograde	46	0.97	0.02	0.157401	0.023208
	Retrograde	46	0.82	0.02	0.146043	0.021533
AICDΔNPTY	Anterograde	76	1.50	0.02	0.125715	0.014421
	Retrograde	76	1.14	0.01	0.122468	0.014048
AICD + PTB2	Anterograde	25	1.40	0.02	0.165501	0.033100
	Retrograde	27	1.19	0.02	0.123390	0.023746

## Data Availability

Data will be made available upon request.

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
