# Peer review of "Axon-Autonomous Effects of the Amyloid Precursor Protein Intracellular Domain (AICD) on Kinase Signaling and Fast Axonal Transport"

_cells, 2023, doi:10.3390/cells12192403_

Round 1

Reviewer 1 Report

The study by Konig et al. revealed that AICD inhibits both anterograde and retrograde FAT. This effect is linked to abnormal activation of axonal kinases and increased phosphorylation of conventional kinesin, the main motor protein for anterograde FAT. Interactions between AICD and axonal proteins containing PTB domains play a pivotal role in this mechanism. These findings suggest a potential contribution of AICD-induced FAT impairment to AD pathogenesis. This proof of concept study is based solely on experiments performed using squid axoplasm. Further studies are required in vitro and in vivo to validate these findings.

Comments

Pg1 line 83: Write full form for PTB.

Figure 2. Total KHC or GST ab would be a better loading control; the authors do not indicate which phosphorylated protein was used as a loading control.

Figure 3B. This blot should present the total forms of the analyzed phosphorylated proteins.

Reviewer 2 Report

The article is devoted to the study of the molecular mechanism of the influence of AICD on fast axonal transport. The experiments were carried out on the high level, results are described very clear, and the obtained results are in line with literature datas. The article is neede only minor clarifications.

1. Since the authors have quantitatively analysed the effect AICD and its mutant forms on FAT, it's important to ensure that concentration and quality of preparation are stable. The distribution of the whole GST-AICD and its fragments is different for of AICD and its mutant forms- Fig. 5B. The distribution of the whole GST-AICD and its fragments is different for of AICD o the Fig 1B and Fig 5B. 

Please add the storage conditions of proteins in the points 2.1 and 2.2.

Please indicate how you calculate the concentration of AICD and its mutant forms depending on the state of preparation.

2. The figure caption (Fig 3C) indicates that the results were normalized to that from anti-KHC-antibody. What about results on the Fid 2C? The bands of controls are differs. Were the results normalized to account for this difference? If yes, please indicate it.

3. Fig 5C, Fig 5D and Table 1. The mutant forms AICDΔPEER and AICDΔYTSI have stronger effect on FAT compared to AICD. Please comment it.

4. Fig 4B. Concentration of SP600125 is different on the Fig, in the figure caption and in the text. Please correct it.

5. Dashed lines on the Fig 4E, 5F and 6D are shifted. Please correct them.
